# *Phyllanthus niruri* Linn.: Antibacterial Activity, Phytochemistry, and Enhanced Antibiotic Combinatorial Strategies

**DOI:** 10.3390/antibiotics13070654

**Published:** 2024-07-16

**Authors:** Gagan Tiwana, Ian E. Cock, Matthew J. Cheesman

**Affiliations:** 1School of Pharmacy and Medical Sciences, Gold Coast Campus, Griffith University, Gold Coast 4222, Australia; g.tiwana@griffith.edu.au; 2School of Environment and Science, Nathan Campus, Griffith University, Brisbane 4111, Australia; i.cock@griffith.edu.au

**Keywords:** antimicrobial resistance, combinational therapies, MRSA, metabolomics, flavonoids, plant extracts

## Abstract

Antimicrobial resistance (AMR) is a global public health threat caused by the misuse and overuse of antibiotics. It leads to infections becoming difficult to treat, causing serious illness, disability, and death. Current antibiotic development is slow, with only 25% of current antibiotics exhibiting novel mechanisms against critical pathogens. Traditional medicinal plants’ secondary metabolites offer potential for developing novel antibacterial compounds. These compounds, often with strong antimicrobial activity, can be used to develop safe and effective antibacterial chemotherapies. This study investigated the antibacterial activity of *Phyllanthus niruri* Linn. extracts against a panel of bacterial pathogens using disc diffusion and microdilution assays and quantified by calculation of minimum inhibition concentration (MIC). Additionally, the effects of combinations of the extracts and selected conventional antibiotics were examined by sum of fractional inhibition concentration (ƩFIC) calculation and isobologram analysis. Liquid chromatography–mass spectrometry (LC-MS) phytochemistry analysis was used to identify noteworthy compounds in the active extracts and the *Artemia* nauplii bioassay was used to evaluate toxicity. The aqueous and methanolic extracts exhibited notable antibacterial activity in the broth microdilution assay against *Staphylococcus aureus* and methicillin-resistant *S. aureus* (MRSA) (MIC = 669 µg/mL and 738 µg/mL, respectively). The methanolic extract also showed noteworthy antibacterial action in the broth assay against *Klebsiella pneumoniae* (MIC = 738 µg/mL). The aqueous extract had noteworthy growth inhibitory activity against *Bacillus cereus* (MIC = 669 µg/mL), whilst the methanolic extract demonstrated good antibacterial activity against that bacterium (MIC = 184 µg/mL). The aqueous and methanol extracts showed minimal antibacterial action against *Shigella flexneri* and *Shigella sonnei*. The extracts were subjected to LC-MS analysis, which revealed several interesting phytochemicals, including a variety of flavonoids and tannins. The antibacterial activity and lack of toxicity of the *P. niruri* extracts indicates that they may be worthwhile targets for antibiotic development and further mechanistic and phytochemistry studies are required.

## 1. Introduction

Antimicrobial resistance (AMR) arises when bacteria, fungi, parasites, or viruses become less susceptible to specific conventional antibiotics [1]. AMR, a natural process in pathogens, occurs when genetic material changes over time. The misuse and overuse of antibiotics in humans, animals, and plants has accelerated the development of antibiotic-resistant pathogens [1]. Due to AMR, infections become difficult to treat, which leads to serious illness and morbidities, disability, and even death. Indeed, The World Health Organisation (WHO) classes AMR as one of the most serious global public health threats, which was directly responsible for 1.27 million deaths worldwide in 2019 [2]. Furthermore, the World Bank reported that AMR has a significant economic burden on the healthcare system, with an estimated cost of USD 1 trillion by 2050 [3]. In particular, the 2022 Global Antimicrobial Resistance and Use Surveillance System (GLASS) study reported that third-generation cephalosporin-resistant *Escherichia coli* and methicillin-resistant *Staphylococcus aureus* had very high rates of resistance and accounted for substantial suffering globally [4]. That report also correlated the high global use of β-lactams/cephalosporins (ceftriaxone and ceftazidime) and fluoroquinolones (ciprofloxacin and levofloxacin) with substantial increases in the incidence of AMR *E. coli* and *Klebsiella pneumoniae* infections [4]. Another study highlighted the development of resistance mechanisms towards vancomycin, daptomycin, ceftaroline, and linezolid in *S. aureus* [5]. Pakbin et al. (2021), reported substantially increased rates of multi-drug-resistant *Shigella* species infections that are now relatively unaffected by conventional antibiotics, including sulfamethoxazole/trimethoprim (83%), amoxicillin (67%), streptomycin (67%), tetracycline (61%), ampicillin (50%), amoxicillin–clavulanic acid (50%), azithromycin (50%), and chloramphenicol (50%) [6].

Unfortunately, the development of new antibiotics is slow, and <25% of current antimicrobials in the development pipeline represent novel mechanisms [7]. Additionally, none of the novel compounds highlighted in that report are active against the WHO critical threat pathogens *Enterococcus faecium*, *S. aureus*, *K. pneumoniae*, *Acinetobacter baumannii*, *Pseudomonas aeruginosa*, and *Enterobacter* species (ESKAPE). A recent report by the WHO showed a rise in antibacterial agents from 80 in 2021 to 93 in 2023, although those therapies lacked novel mechanisms that target critical pathogens. Only two of the thirteen approved antibiotics since 2017 represent a new chemical class [8].

The present study focuses on the antibacterial properties and phytochemistry profile of the Ayurvedic medicinal plant *Phyllanthus niruri* Linn. Leaf extracts of this plant were tested against a panel of bacterial pathogens important to human health. The species tested included *S. aureus*, methicillin-resistant *S. aureus* (MRSA), *E. coli*, extended spectrum β-lactamase (ESBL) *E. coli*, *K. pneumoniae*, ESBL *K. pneumoniae*, *Salmonella typhimurium*, *Shigella sonnei*, *Shigella flexneri*, and *Bacillus cereus*. Traditionally, dried *P. niruri* leaf powder is used in the form of infusions and decoctions to treat dysentery and diarrhoea [9]. Literature evidence shows that the *P. niruri* extracts have antiviral, antibacterial, hypolipidemic, hypoglycaemic, analgesic, anti-inflammatory, and cardioprotective properties because of their novel bioactive compounds [10]. Amin et al. (2012) demonstrated the antibacterial effects of a 100 mg/mL aqueous extract of the entire *P. niruri* plant against *S. aureus* and *Streptococcus agalactiae*. The disc diffusion method was used to calculate the zones of inhibition (ZOIs) for *S. agalactiae* and *S. aureus*, which were 12 and 20 mm, respectively [11]. However, the minimum inhibition concentration (MIC) values were not determined using conventional broth microdilution techniques. Instead, they were observed based on the absence of turbidity in bacteria that were incubated overnight with broth-diluted plant extract and, therefore, the results of that study are difficult to compare with other studies using more conventional methods. Furthermore, the previous study did not include any strains of *S. aureus* that were resistant to antibiotic treatment [11]. Ibrahim and colleagues [12] prepared methanolic extracts of *P. niruri*; however, the process of extraction yielded an oily paste, which may have contained traces of methanol solvent. Furthermore, activity on agar was only possible at very high concentrations (100 mg/mL) and high MIC values were determined using microdilution broth assays for *B. cereus*, *B. subtilis*, and *S. aureus* (3.13–6.25 mg/mL), as well as for *E. coli*, *P. aeruginosa*, and *P. rettgeri* (25 mg/mL).

A separate investigation demonstrated the antibacterial properties of an aqueous extract from *P. niruri* leaves using the disc diffusion well method against *E. coli* and *P. aeruginosa* [13]. The inhibitory effects of the plant extract against those pathogens were observed when a high volume (1000 µL) and concentration (25 mg/mL) of extract were used. Furthermore, the method of plant extraction was inadequately specified, and no minimum inhibitory concentration (MIC) values were established. Another study reported antibacterial activity of an aqueous *P. niruri* whole plant extract against *S. aureus*, *E. coli*, and *S. typhi*, with MIC values ranging between 15 mg/mL and 62 mg/mL using disc diffusion assays [14]. Notably, these MIC values are very high and would normally be classified as indicating a lack of activity. These MIC values were not confirmed using broth microdilution assays, and, therefore, comparisons with other studies are not possible.

The current study investigates the antibacterial activity of *P. niruri* extracts on a panel of bacterial pathogens, including antibiotic-resistant strains, using disc diffusion and liquid microdilution assays. Liquid chromatography–mass spectrometry (LC-MS) was used to investigate the phytochemical profile of the *P. niruri* extracts and highlight noteworthy compounds. Additionally, our study examined the interactions between active plant extracts and a wide range of reference antibiotics when tested in combination. *Artemia franciscana* Kellogg nauplii lethality assays were also conducted to investigate the toxicities of all extracts.

## 2. Results

### 2.1. Antimicrobial Susceptibility Assays

Individual 1 g masses of *P. niruri* leaf powder were extracted separately using either water, methanol, or ethyl acetate as extract solvents, followed by drying and resuspension in 10 mL of 1% DMSO. The resulting concentrations of the extracts were 10.7, 11.8, and 8.1 mg/mL, respectively. The antibacterial activities of all extracts and reference antibiotics were assessed using both disc diffusion (measured as ZOI values) and broth microdilution assays and expressed as MIC values. The aqueous and methanolic extracts exhibited low to good antibacterial activity against seven bacterial pathogens in the disc diffusion and broth dilution assays (Figure 1 and Figure 2, and Table 1). Noteworthy antibacterial activity was observed for the aqueous and methanolic extracts in the broth microdilution assay against *S. aureus* and MRSA (MIC = 669 µg/mL and 738 µg/mL, respectively). Similar results were also noted for both extracts against *S. aureus* and MRSA in disc diffusion assay. In contrast, the methanolic extract showed low antibacterial activity in the broth microdilution assay against ESBL *E. coli,* with MIC = 2950 µg/mL. Interestingly, no antibacterial activity in broth was observed against the antibiotic-sensitive *E. coli* strain by any extract, despite significant inhibition noted for the methanol extract against *E. coli* on agar in the disc diffusion assay. The methanolic extract exhibited notable antibacterial activity against *K. pneumoniae* in broth, with an MIC of 738 µg/mL.

In contrast, the aqueous extract also exhibited moderate antibacterial activity against *K. pneumoniae* (MIC = 1338 µg/mL). However, no inhibitory activity was evident for the aqueous and methanolic extracts against *K. pneumoniae* in the solid-phase agar plate assays. Notably, none of the extracts inhibited the growth of ESBL *K. pneumoniae* or *S. typhimurium* in either the disc diffusion or broth dilution assays. Low antibacterial activity was observed for the aqueous and methanolic extracts against *S. sonnei* and *S. flexneri* in both assays. The methanolic extract showed good antibacterial activity against *B. cereus* (MIC = 184 µg/mL), whilst the aqueous extract exhibited noteworthy antibacterial activity against that bacterium (MIC = 669 µg/mL). The ethyl acetate extract did not exhibit any antibacterial activity on agar diffusion or broth dilution assays against any of the bacterial pathogens tested in this study.

### 2.2. Combination Assays—Fractional Inhibitory Concentration (FIC) Determinations

Combinations of the *P. niruri* extracts and conventional antibiotics were tested for possible interactions against both antibiotic-sensitive and resistant bacterial strains (Table 2). No synergistic interactions were observed for any combination. Twenty combinations had additive effects, and thirty-seven combinations were indifferent. Additionally, twelve combinations had antagonistic effects.

### 2.3. Identification of Compounds in the P. niruri Aqueous, Methanol, and Ethyl Acetate Extracts

Due to their greater antibacterial activity, the methanolic and aqueous extracts were deemed to be the most promising for phytochemical investigation. The metabolomic fingerprints of all extracts were examined using LC-MS fingerprinting analysis, with an emphasis on the flavonoid, tannin, and terpenoid components. Most of the extract compounds eluted in the isocratic stage of 30% to 90% acetonitrile (Appendix A), indicating that most of the extract components were relatively high polarity [15]. Organic acids and amines favour polar environments and elute early in the chromatogram. In contrast, lipophilic substances and hydrocarbons, which have a stronger interaction with the non-polar stationary phase, elute later as the gradient continues. Only those compounds that were totally matched in any of the databases (and were not present in the blank controls) were selected to make a putative list of all the compounds identified (Appendix A). Herein, we have focussed on the flavonoid and tannin compounds (Table 3).

### 2.4. Toxicity Quantification

*Artemia franciscana* Kellogg nauplii lethality assays (ALA) were used to evaluate the toxicity of the plant extracts in triplicate on 48-well plates to determine their level of toxicity. After a 24 h exposure, plant extracts were considered hazardous if they had LC_50_ values <1000 µg/mL [16]. After this duration of incubation, more than half of the nauplii tested in this study were still alive for all extracts, and all plant extracts tested were therefore classified as nontoxic. Indeed, the results of all extracts were comparable to those of the negative control (artificial sea water).

## 3. Discussion

This study screened *P. niruri* leaf extracts against a panel of bacterial pathogens important to human health, including antibiotic-resistant strains. Notably, both the aqueous and methanolic extracts inhibited the growth of seven of the ten bacterial pathogens tested, highlighting their potential as antibiotic development targets. The methanolic extracts demonstrated the greatest antibacterial activity in both the disc diffusion and liquid microdilution assays. In contrast, ethyl acetate extracts did not inhibit the growth of any of the bacterial strains tested in this study. These differences may be due to the varying net yields and types of phytochemicals extracted by different solvents. Methanol and water, being more polar, extract an abundance of high- to mid-polarity phytochemicals, whilst ethyl acetate extracts fewer (and lower-polarity) compounds [17]. These differences in phytochemical compositions between the extracts may also contribute to the different apparent antibacterial growth inhibition effects between the liquid dilution and disc diffusion studies. Less-polar or larger-sized phytochemicals do not penetrate through solid agar, thereby reducing their antibacterial activity. The polarity of these phytochemicals also affects their solubility in broth, which, in turn, influences their capacity to dissolve and may result in incorrect MIC values [18]. A previous study has provided evidence that agar depth in the petri dishes as well as agar uniformity can affect the size of ZOIs in the agar disc diffusion assays [19]. Although we prepared uniform agar according to the manufacturers’ instructions and poured the agar at a consistent depth of 4 mm, we acknowledge that the disc diffusion assays should be repeated at different agar depths in order to verify the ZOIs of all of the extracts and antibiotics. Additionally, while the ZOI boundaries were clearly visible in our study, it should be noted that staining plates with methylene blue or crystal violet [20] may further enhance the visibility of ZOIs should lack of clarity in this regard occur in future studies.

In our study, MRSA exhibited resistance to the reference antibiotics, including β-lactams (penicillin G, oxacillin, amoxicillin), and macrolides (erythromycin). The significance of MRSA’s resistance to β-lactams and macrolides is noteworthy [21]. β-lactam antibiotics have been extensively used due to their efficacy in treating various infections and their well-established safety profiles [4,5]. However, the production of β-lactamase enzymes has rendered these medicines ineffective against some bacterial strains. Similarly, macrolide antibiotics, such as erythromycin, are widely used due to their capacity to hinder protein synthesis in bacteria [21]. The presence of MRSA resistance to macrolides makes treatment alternatives difficult, emphasising the urgent requirement for new therapeutic agents and techniques. Therefore, it is crucial to discover novel compounds that can evade or overcome these resistance mechanisms. The aqueous and methanol *P. niruri* extracts exhibited noteworthy antibacterial activity against *S. aureus* and MRSA, with similar MIC values (669 µg/mL and 738 µg/mL, respectively). This suggests that the resistance mechanisms found in the MRSA strain have minimal impact on the active components of the extracts. Therefore, the extract components either operate through alternative mechanisms or hinder the bacterial resistance pathways. The presence of the *mecA* gene is the most important factor in MRSA resistance [22]. This gene encodes a novel penicillin-binding protein (PBP2a), which possesses a diminished affinity for β-lactam antibiotics. This protein enables the bacteria to persist by allowing them to synthesise cell wall despite the presence of these antibiotics. Therefore, the processes by which the extract phytochemicals work may differ from those of β-lactam antibiotics, even in β-lactam-resistant bacteria. In contrast, these extracts may include active compounds that can interfere with the bacterial strains’ defence systems against these antibiotics [22]. This finding is encouraging because, in comparison to their susceptible counterpart, the MRSA strain examined in this study showed significantly reduced susceptibilities (i.e., greater resistance) to a variety of antibiotics from various classes, including β-lactams, macrolides, fluoroquinolones, aminoglycosides, and tetracyclines.

The methanol extract of *P. niruri* demonstrated noteworthy antibacterial activity against *K. pneumoniae* (MIC = 738 µg/mL), although no activity was noted against the corresponding antibiotic-resistant strain ESBL *K. pneumoniae*. The resistance of ESBL *K. pneumoniae* against the plant extracts may be attributed to the presence of enhanced β-lactamase enzymes, increased efflux mechanisms, altered cell wall permeability, biofilm formation, and genetic resistance factors [23]. ESBL-producing bacteria may also have a diversity of other resistance mechanisms that limit the efficacy of many antimicrobial compounds, including those produced from medicinal plants [24].

Both aqueous and methanol *P. niruri* extracts exhibited noteworthy to good antibacterial activity against *B. cereus* (MIC = 669 µg/mL and 184 µg/mL, respectively). The phytochemicals present in both extracts may be specifically affecting the structure and functions of *B. cereus*, which includes breakdown of cell membranes, inhibition of vital enzymes, interference with DNA synthesis, and other crucial processes necessary for the survival of bacteria [25]. Additionally, the presence of numerous phytochemicals in the extracts amplifies their overall antibacterial effectiveness [9].

Our study also investigated the utilisation of *P. niruri* extracts in conjunction with conventional antibiotics. This strategy has particular promise for the development of new antibiotic chemotherapies, as numerous bacteria have acquired resistance to current antibiotics [4]. Our objective was to augment the effectiveness of antibiotics and potentially overcome resistance mechanisms by combining them with plant extracts. Augmentin^®^, a combination of amoxicillin and clavulanic acid, is a prominent example of how combining medicines can enhance therapeutic outcomes as a combination therapy [26]. Using clavulanic acid, Augmentin^®^ inhibits the β-lactamase enzymes produced by resistant bacteria. This makes amoxicillin able to target and eradicate the bacteria effectively. Clavulanic acid functions as a β-lactamase inhibitor, preventing the degradation of the antibiotic by binding irreversibly to the enzyme’s active site. In the absence of this mechanism, amoxicillin would be ineffective against bacteria that generate ESBL enzymes.

Similarly, the observed additive effects between penicillin G, amoxicillin, and plant extracts against *S. aureus* and *B. cereus* in our study can be attributed to the presence of phytochemicals in the extracts that possess anti-β-lactamase properties [27]. These phytochemicals may inhibit the β-lactamase enzymes which are responsible for antibiotic resistance by hydrolysing the β-lactam ring present in conventional antibiotics like penicillin and amoxicillin [28]. Additionally, phytochemicals present in the plant extracts may be interacted with the β-lactamase enzymes in a manner similar to clavulanic acid, thereby preserving the antibiotics from enzymatic degradation and enhancing their antimicrobial activity. As a result, plant extracts may have the potential to serve as an alternate therapeutic approach and may be a safe and less harmful strategy for combating antibiotic resistance [29]. Natural compounds found in plants often have fewer adverse effects and a lower probability of generating resistance compared to synthetic antibiotics [9]. Furthermore, the possibility of resistance development may be mitigated by the diversity of phytochemicals in plant extracts, which may target multiple bacterial pathways.

Our study indicated that combination of tetracycline with methanol *P. niruri* extract shows additive interactions against *K. pneumoniae* and *S. flexneri*, indicating improved antibacterial effectiveness. The resistance to tetracycline is commonly caused by tetracycline-specific efflux pumps [30]. Therefore, the observed additive effect suggests that the plant extract may have inhibited these efflux pumps. The suppression of efflux pumps enables tetracycline to have an extended intracellular presence, hence enhancing its efficacy. Although ribosomal modifications can also lead to tetracycline resistance, this route is less frequent [30]. Our study demonstrates that the methanolic *P. niruri* extract contains a diverse array of phytochemicals that possess broad-spectrum antibacterial properties. Similarly, plant extracts from *Berberis vulgaris* L. and *Piper nigrum* L. also block tetracycline efflux pumps [31]. Therefore, investigating natural phytochemicals as adjuvants is a hopeful strategy for addressing antibiotic-resistant bacteria and improving traditional antibiotic treatments. Additional investigation into these combinations has the potential to yield more efficient therapeutic approaches against multi-drug-resistant pathogens.

Notably, polymyxin B and *P. niruri* extracts (aqueous and methanol) combinations exhibited substantial antagonistic interactions. These interactions may be attributed to the binding of bioactive phytochemicals in the extracts with polymyxin B, which inhibits its absorption and effectiveness in targeting bacterial cells [32]. Furthermore, it should be noted that the efficacy of polymyxin B is influenced by the pH of the environment. It is possible that the introduction of plant extracts into the broth may have caused a change in the local pH, thereby affecting the activity of polymyxin B [33]. The changes in the pH environment lead to diminished binding affinity of polymyxin B to bacterial membranes, thus reducing its effectiveness.

LC-MS metabolomics analysis of the *P. niruri* extracts highlighted the flavonoid, tannin, and terpenoid components (Table 2). A wide array of flavonoids and their derivatives was identified in the methanol *P. niruri* extracts, including epicatechin (Figure 3A), fustin (Figure 3B), orientin (Figure 3C), corymboside (Figure 3D), hyperoside (Figure 3E), quercetin (Figure 3F), vitexin (Figure 3G), miquelianin (Figure 3H), myricitrin (Figure 3I), fisetin (Figure 3J), rutin (Figure 3K), trifolin (Figure 3L), kaempferol (Figure 3M), astragalin (Figure 3N), and apigenin (Figure 3O). Notably, a previous study also documented the presence of similar compounds in *P. niruri* extracts, including rutin, quercetin, nirurin, astragalin, limonene, p-cymene, lupeol, and ellagic acid. However, those previous studies did not investigate antibacterial activity for those *P. niruri* extracts, or the individual compounds. Indeed, our study is the first to identify this diverse range of bioactive phytochemicals in *P. niruri* leaf extracts and link them with antibacterial activity.

Flavonoids and their derivatives are widely recognised for their antimicrobial activity. For example, noteworthy antibacterial activity has been reported for quercetin against five clinical strains of MRSA, with MIC values ranging from 16 to 64 µg/mL [34]. Additionally, dihydromyricetin and myricetin demonstrated strong anti-MRSA activity in the same study (MIC = 10 µg/mL and 25 µg/mL, respectively). Another study reported that flavonoids exert anti-MRSA activity through the modulation of the PBP2a proteins [35]. Buchmann et al. (2022) reported synergistic combinations of myricetin with tetracycline and ciprofloxacin against ESKAPE category *K. pneumoniae* strains [36]. That study demonstrated that the MIC value of tetracycline against *K. pneumoniae* (PBIO1455) was decreased by myricetin from 11 μg/mL to 3.3 μg/mL. Another study demonstrated the synergistic effect of gallic acid and norfloxacin against *S. aureus* [37]. Gallic acid decreased the MIC of norfloxacin from 156 μg/mL to 49 μg/mL. Combining gallic acid with gentamicin lowered the MIC against *S. aureus* from 49 μg/mL to 2.5 μg/mL [37]. The same study found that pyrogallol, in combination with either norfloxacin or gentamicin operated synergistically against *S. aureus*. Further studies are required to screen the extracts against the other ESKAPE bacteria to more fully evaluate their potential against these pathogens.

In our study, the *P. niruri* extracts lacked synergistic interactions with any of the tested antibiotics against all the bacterial pathogens investigated. Phytochemicals in *P. niruri* extracts may have anti-β-lactamase activity, as evidenced by their additive interactions with penicillin-G and amoxycillin against *S. aureus* and *B. cereus*. Further research is needed to investigate the anti-β-lactamase capability of these extracts. Furthermore, the *P. niruri* extracts showed noteworthy broad-spectrum antibacterial activity against *K. pneumoniae, S. aureus*, and MRSA. This is encouraging because these bacterial pathogens are the leading cause of healthcare-associated and community-acquired illnesses [38]. This underlines *P. niruri*’s potential as a valuable source of antimicrobial compounds for future research and development in the fight against bacterial illnesses. The toxicity quantification assay revealed that all extracts of *P. niruri* are nontoxic, thus indicating their safety as an antimicrobial agent. To determine whether these extracts are suitable for use in medicine, additional testing must be conducted on a variety of human cell lines.

## 4. Materials and Methods

### 4.1. Source of Plant Samples

*Phyllanthus niruri* Linn. leaf powder manufactured by Parijata Herbs was purchased online from Navafresh Australia. The traditional Ayurvedic names of *P. niruri* (Bhumi amla, Nelanelli powder) were used to search for the plant herb on the supplier’s website. The authenticity and quality of the plant materials were confirmed by the supplier. Plant material was labeled and voucher specimens stored at the School of Pharmacy and Medical Sciences, Griffith University, Gold Coast campus.

### 4.2. Preparation of Extracts

Individual 1 g masses of *Phyllanthus niruri* Linn. leaf powders were weighed into three tubes and sterile deionized water, methanol (AR grade), and ethyl acetate (AR grade) were added to give a total volume of 50 mL. Organic solvents (methanol and ethyl acetate) were supplied by ChemSupply, Gillman, Australia. Samples were mixed by continuous rolling at 30 rpm for 24 h at room temperature. The samples were then filtered under vacuum pressure through Whatman No. 54 filter paper (Sigma-Aldrich, Melbourne, Australia) into pre-weighed 50 mL tubes. The aqueous samples were freeze-dried by lyophilization for 3 days in an Alpha 1-4 LSCplus benchtop freeze dryer (Martin Christ, Osterode am Harz, Germany). Organic solvent samples were evaporated at 40 °C for 2 days or until evaporation was complete. All dried extracts were weighed to determine the final yields and then resuspended in 10 mL 1% dimethyl sulfoxide (DMSO; Merck, Macquarie Park, Australia) and sonicated three times using 20 s pulses of a probe sonicator set at 1kHz, with 30 s rest between pulses. All extracts were subsequently sterilized by passage through 0.22 µm filters (Sarstedt, Mawson Lakes, Australia) and stored at −20 °C until use.

### 4.3. Antibiotics and Bacterial Strains

Powdered antibiotics were purchased from Sigma-Aldrich (Melbourne, Australia), which included penicillin G (potency of 1440–1680 µg/mg), erythromycin (potency ≥850 µg/mg), tetracycline (≥95% purity by HPLC), chloramphenicol (≥98% purity by HPLC), ciprofloxacin (≥98% purity by HPLC), polymyxin B (purity >90%), oxacillin (≥95% purity by TLC), amoxycillin (potency of 900 µg/mg), gentamicin (≥98% purity by HPLC), and vancomycin (potency of ≥900 μg per mg). Antibiotic stock (1 mg/mL) solutions were prepared for broth microdilution assays and stored at −20 °C until use. Preloaded standard discs containing penicillin G (10 IU), erythromycin (10 µg), tetracycline (30 µg), chloramphenicol (30 µg), ciprofloxacin (1 µg), polymyxin B (300IU), oxacillin (1 µg), gentamicin (10 µg), vancomycin (30 µg), cefoxitin (30 µg), and Augmentin^®^ (15 µg) were purchased from Oxoid Ltd., Thebarton, Australia. A volume of 10 µL of amoxycillin stock solution (0.01 mg/mL) was infused into sterile filter paper discs and placed on Mueller–Hinton (MH) agar and used immediately for disc diffusion assays. All reference antibiotics except Augmentin^®^ and cefoxitin were also used in broth microdilution assays.

Reference strains of *Escherichia coli* (ATCC 25922), *Staphylococcus aureus* (ATCC 25923), MRSA (ATCC 43300), *Klebsiella pneumoniae* (ATCC 13883), and ESBL *Klebsiella pneumoniae* (ATCC 700603), *Shigella sonnei* (ATCC 25931), *Salmonella typhimurium* (Salmonella enterica serovar Typhimurium; ATCC 14028), *Shigella flexneri* (ATCC 12022), and *Bacillus cereus* (ATCC 14579) were obtained from the American Type Culture Collection (ATCC, Manassas, VA, USA). A clinical isolate strain of ESBL *Escherichia coli* strain was acquired from the Gold Coast University Hospital, Southport, QLD, Australia. All bacterial strains were grown in Mueller–Hinton (MH) agar and broth (Oxoid Ltd., Australia). The MRSA strain was cultured at 35 °C to maintain its resistance phenotype [39], whilst all other bacterial strains were cultured at 37 °C for 18–24 h.

### 4.4. Antibacterial Susceptibility Screening

A modified Kirby–Bauer disc diffusion method was used to investigate the antibacterial activity of all plant extracts in MH agar [16]. Briefly, all bacterial strains were inoculated in 40 mL MH broth using individual colonies isolated from MH agar plates grown at 37 °C (with the exception of MRSA, which was grown at 35 °C) for 18–24 h. The individual bacterial cultures were then used to make 0.5 McFarland standards for each strain and 100 µL of the cultures was spread on fresh MH agar plates. Whatman sterile filter paper discs (6 mm diameter) were affixed onto MH agar with sterile forceps and 10 µL of all extracts resuspended in 1% DMSO infused into them. All reference antibiotics discs were placed on MH agar inoculated with each bacterial strain. All samples were tested in triplicate. The extract and reference antibiotic plates were incubated for 18–24 h at 37 °C (except MRSA, which was incubated at 35 °C).

The diameter of the inhibition zones around each disc was measured to the closest whole mm to evaluate the bacterial growth inhibition and reported as the zone of inhibition (ZOI). Samples with no antibacterial activity were reported with ZOI of 6 mm (the diameter of the discs). Mean values (±SEM) are reported in this study. The ZOI data are presented in the form of bar graphs as the average ± SEM of a minimum of three independent studies. Differences between the negative controls and the treatment groups were analysed using one-way analysis of variance (ANOVA), with *p*-values < 0.01 and <0.001 considered to be highly statistically significant and very highly significant, respectively. In the bar graphs, *p*-values < 0.01 are represented with a single asterisk symbol (*), while *p*-values < 0.001 are represented with a double asterisk symbol (**).

### 4.5. Minimum Inhibitory Concentration (MIC)

MIC values for all extracts and reference antibiotics were quantified using a standard 96-well microtitre plate broth microdilution assay [40]. Individual 100 µL volumes of all extracts and reference antibiotics were added into the top row of the plates and serially diluted down each column of the plates using doubling dilutions. A volume of 100 µL of 1:100 dilution of a 0.5 McFarland cell suspension was then added to all wells (except the sterile controls) and incubated at 37 °C for 20–24 h. Following this incubation, 40 μL of p-iodonitrotetrazolium violet (INTZ; Sigma Aldrich, Australia) dye solution (0.4 mg/mL) was added into all wells on the plate and re-incubated for an additional 2–4 h at room temperature. Visual inspection was used to determine the MIC values as the lowest concentration of plant extracts (or antibiotics) that completely inhibited bacterial growth (visualised as lack of a red–pink colour change). All experiments were performed in duplicate. Extract MIC values > 10,000 μg/mL were considered to be inactive; MICs between 2000 and 10,000 μg/mL were defined as low activity; MIC values between 1000 and 2000 μg/mL as moderate activity; 400–1000 μg/mL as noteworthy activity; 100–400 μg/mL was defined as good activity; and MIC values < 100 μg/mL were deemed to have high activity.

### 4.6. Fractional Inhibitory Concentration (FIC) Evaluation

The plant extracts and reference antibiotics that showed antibacterial activity (minimum inhibitory concentration (MIC) ≤ 3000 µg/mL and ≤2.5 µg/mL, respectively) were further chosen to conduct 50:50 ratios to examine their interactions with susceptible bacterial pathogens. The interactions between the extracts and antibiotics were investigated by determination of the sum of the fractional inhibitory concentrations (ΣFIC) for each combination using the following equations (a = extracts; b = antibiotics):FIC(a) = MIC (a in combination with b)/MIC (a independently)
FIC(b) = MIC (b in combination with a)/MIC (b independently).

∑FIC was then calculated using the formula ∑FIC = FIC(a) + FIC(b).

The interactions were classified as synergistic (∑FIC ≤ 0.5), additive (∑FIC > 0.5–≤ 1.0), indifferent (∑FIC > 1.0–≤ 4.0), or antagonistic (∑FIC > 4.0) [41].

### 4.7. Toxicity Assays

The toxicities of the plant extracts and controls were investigated using standard *Artemia franciscana* Kellogg nauplii lethality assays (ALA) [42]. Briefly, 400 µL of plant extracts (2mg/mL diluted in artificial seawater) and 400 µL of artificial seawater containing newly hatched (within 1 day) *A. franciscana* nauplii were dispensed into individual wells of a 48-well plate. A 400 µL volume of negative control (32 g/L artificial seawater; Red Sea) and positive control (1 mg/mL sodium azide) was included on all plates and tested in parallel. The plates were then incubated for 24 h at 25 °C ± 1 °C and then the live shrimps were counted. Probit analysis was used to calculate LC_50_ values as the concentration of extract or control to kill 50% *A. franciscana* nauplii in individual wells. The mean percentage of three repeated experiments was used to calculate LC_50_ values graphically as the concentration that induced 50% mortality.

### 4.8. Non-Targeted Headspace LC-MS Workflow for Quantitative Analysis

Non-targeted headspace metabolic profile analysis of all extracts was performed using a Vanquish Ultra High-Performance Liquid Chromatography (UHPLC) system (Thermo Fisher Scientific, Waltham, MA, USA). Compounds separation used an Accucore ™ RP-MS column (100 mm × 2.1 mm) with particle size of 2.6 μm, connected to an Orbitrap Exploris 120 mass spectrometer (Thermo Fisher Scientific). The UHPLC system is equipped with a quaternary pump operating at a flow rate of 0.6 mL/min. The pump is coupled to the system and utilises the following mobile phases: (A) 0.1% *v/v* formic acid in ultrapure water and (B) acetonitrile (MeCN) containing 0.1% *v/v* formic acid. Xcalibur acquisition software (version 2.0) connected with the system was used to develop the following gradient flow (total 24 min) for compound separation: 5% B for 5 min, 5% to 30% B for 5 min, 30% B for 3 min, 30% to 90% B for 4 min, isocratic elution at 90% B for 4 min to flush the column, and 90% to 5% B for 1 min. The column was re-equilibrated by running isocratically at 5% B of mobile phase for 2 min between each separation.

The mass spectra of the eluted chemicals were analysed using the Orbitrap Exploris 120 mass spectrometer in information-dependent acquisition (IDA) mode. The Orbitrap system used electrospray ionisation (ESI) in negative ionisation mode, with the following predetermined parameters: vaporiser temperature = 350 °C, sheath gas = 60 psi, auxiliary gas = 15 psi, sweep gas = 2 psi, and spray voltage = 2.5 KV to.5 KV. Compound Discoverer^TM^ software 3.3 was used to examine the data. The natural product unknown ID procedure was chosen, along with local database searches and stats online. To eliminate the backdrop and distinguish distinct extract components, the data files for each extract were examined separately and compared to the blank file. Applying result filters in the Compound Discoverer, databases such as Mz cloud, ChemSpider, Predicted Compositions, and MassList were used to obtain a partial or complete match of the compounds that were found. The Compound Discoverer’s Mz cloud and ChemSpider database must pass communication tests to verify online access, accurately identify eluted chemicals, and prevent workflow mistakes. Using the Compound Discoverer software 3.3, Excel files of the compounds that were identified from each extract were created. These files were then compared with one another to create a list of potential putative compounds. After combining duplicates using pivot table analysis, relative abundance (as a percentage of the total area) was determined.

## 5. Conclusions

The urgent need for new antibacterial medicines has sparked interest in natural products as possible sources. Our findings indicate that active extracts of *Phyllanthus niruri* L. can inhibit the growth of resistant bacteria just as efficiently as susceptible strains, indicating distinct antibacterial mechanisms for the extracts. Furthermore, the methanolic and aqueous extracts potentiate the activity of several conventional antibiotics, particularly, β-lactam antibiotics. This may allow these antibiotics to be reactivated, even in bacterial strains otherwise resistant to their effects. The potentiation mechanism is yet to be determined, but it is possible that extract components may inactivate bacterial extended spectrum β-lactamase enzymes, thereby increasing the intracellular antibiotic concentration. Some of the phytochemicals identified are likely to contribute to these activities, implying that they could be useful as novel antibacterial agents, although this needs to be verified. Future research should focus on isolating these molecules and investigating their potential to enhance the efficacy of current antibiotics. The use of additional methods, such as GC-MS, are needed to identify lower-polarity and volatile molecules, whilst NMR is required to definitively structurally identify the extract components.

## Figures and Tables

**Figure 1 antibiotics-13-00654-f001:**
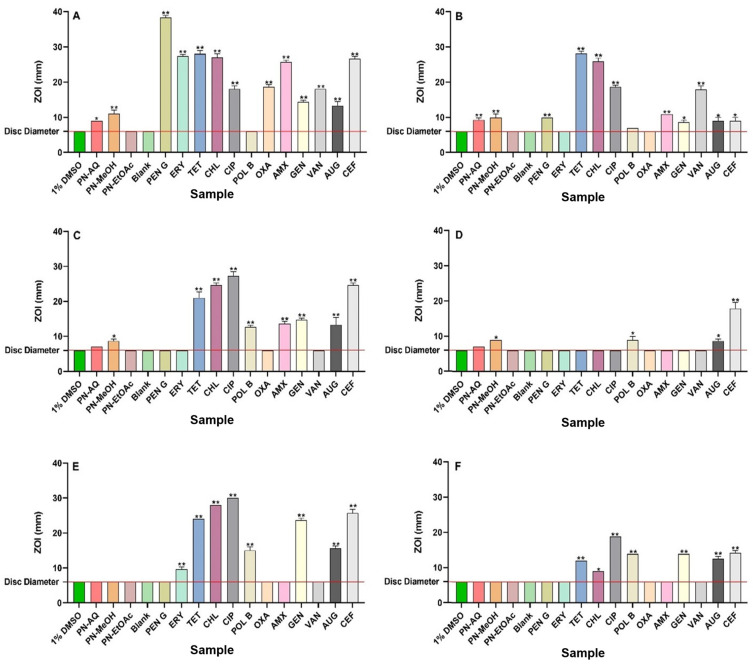
Antimicrobial activity of *P. niruri* leaf extracts in the disc diffusion assays against (**A**) *S. aureus*, (**B**) MRSA, (**C**) *E. coli*, (**D**) ESBL *E. coli*, (**E**) *K. pneumoniae*, (**F**) ESBL *K. pneumoniae*. PN-AQ = *P. niruri* aqueous; PN-MeOH = *P. niruri* methanol; PN-EtOAc = *P. niruri* ethyl acetate. Negative controls = 1% DMSO and Blank = sterile water. Reference antibiotics: PEN G = penicillin G, ERY = erythromycin, TET = tetracycline, CHL = chloramphenicol, CIP = ciprofloxacin, POL B = polymyxin B, OXA = oxacillin, AMX = amoxycillin, GEN = gentamicin, VAN = vancomycin, AUG = Augmentin^®^, CEF = cefoxitin. *X*-axis represents samples (extracts, antibiotics, and negative controls). Horizontal red line on the *y*-axis at 6 mm indicates the disc diameter used in the assay. Mean values (±SEM) are reported from three independent studies. *p*-values < 0.01 are represented with a single asterisk symbol (*), while *p*-values < 0.001 are represented with a double asterisk symbol (**).

**Figure 2 antibiotics-13-00654-f002:**
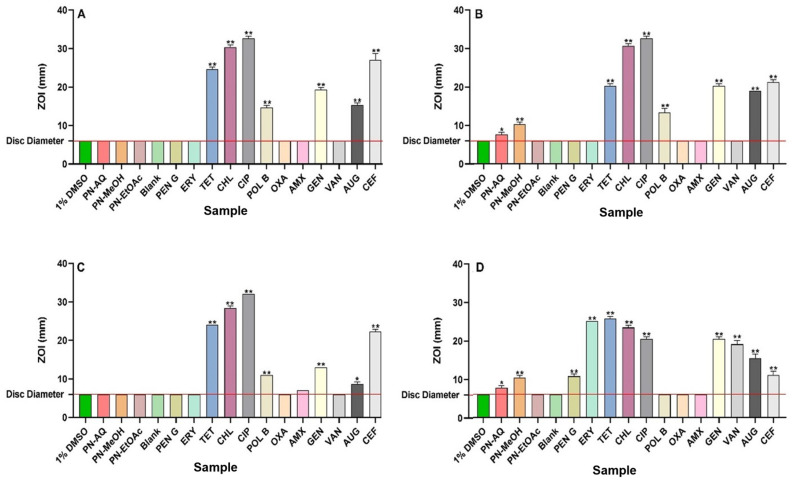
Antimicrobial activity of *P. niruri* leaf extracts in disc diffusion assays against (**A**) *S. sonnei*, (**B**) *S. flexneri*, (**C**) *S. typhimurium*, (**D**) *B. cereus.* PN-AQ = *P. niruri* aqueous; PN-MeOH = *P. niruri* methanol; PN-EtOAc = *P. niruri* ethyl acetate. Negative controls = 1% DMSO and Blank = sterile water. Reference antibiotics: PEN G = penicillin G, ERY = erythromycin, TET = tetracycline, CHL = chloramphenicol, CIP = ciprofloxacin, POL B = polymyxin B, OXA = oxacillin, AMX = amoxycillin, GEN = gentamicin, VAN = vancomycin, AUG = Augmentin^®^, CEF = cefoxitin. *X*-axis represents samples (extracts, antibiotics, and negative controls). Horizontal red line on the *y*-axis at 6 mm indicates the disc diameter used in the assay. Mean values (±SEM) are reported from three independent studies. *p*-values < 0.01 are represented with a single asterisk symbol (*), while *p*-values < 0.001 are represented with a double asterisk symbol (**).

**Figure 3 antibiotics-13-00654-f003:**
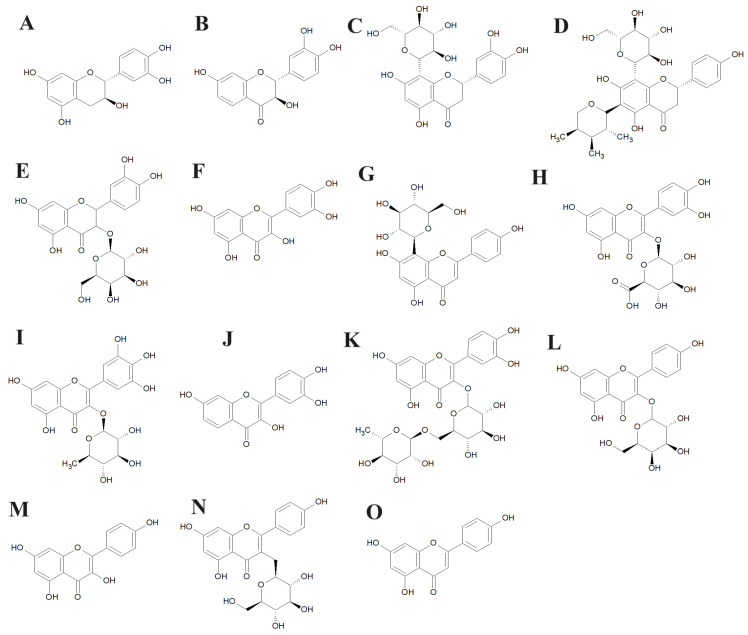
Structures of noteworthy compounds identified in the *P. niruri* leaf extracts: (**A**) epicatechin; (**B**) fustin; (**C**) orientin; (**D**) corymboside; (**E**) hyperoside; (**F**) quercetin; (**G**) vitexin; (**H**) miquelianin; (**I**) myricitrin; (**J**) fisetin; (**K**) rutin; (**L**) trifolin; (**M**) kaempferol; (**N**) astragalin; and (**O**) apigenin.

**Table 1 antibiotics-13-00654-t001:** MIC values (µg/mL) for aqueous and methanol *P. niruri* extracts and the reference (positive control) antibiotics against the ten bacterial strains.

Extract Type or Antibiotic	Bacterial Species & MIC (µg/mL)									
	*S. aureus*	MRSA	*E. coli*	ESBL *E. coli*	*K. pneumoniae*	ESBL *K. pneumoniae*	*S. typhimurium*	*S. sonnei*	*S. flexneri*	*B. cereus*
Aqueous	669	669	Inactive	Inactive	1338	Inactive	Inactive	**2675**	**2675**	669
Methanol	738	738	Inactive	**2950**	738	Inactive	Inactive	**2950**	**2950**	184
Ethyl acetate	Inactive	Inactive	Inactive	Inactive	Inactive	Inactive	Inactive	Inactive	Inactive	Inactive
PEN G	1.25	>2.5	>2.5	>2.5	>2.5	>2.5	>2.5	>2.5	>2.5	0.625
ERY	0.31	>2.5	>2.5	>2.5	>2.5	>2.5	>2.5	>2.5	>2.5	0.08
TET	0.16	0.04	0.31	>2.5	0.625	>2.5	0.625	0.31	0.625	0.08
CHL	>2.5	>2.5	2.5	>2.5	2.5	>2.5	2.5	2.5	1.25	1.25
CIP	0.16	0.625	0.02	>2.5	0.02	0.16	0.02	0.02	0.02	0.08
POLB	>2.5	>2.5	0.02	0.02	0.02	0.04	0.31	0.31	0.31	>2.5
OXA	0.16	>2.5	>2.5	>2.5	>2.5	>2.5	>2.5	>2.5	>2.5	1.25
AMX	0.625	>2.5	>2.5	>2.5	>2.5	>2.5	>2.5	>2.5	>2.5	0.625
GEN	>2.5	>2.5	0.625	>2.5	0.625	>2.5	0.625	2.5	0.625	0.16
VAN	1.25	1.25	>2.5	>2.5	>2.5	>2.5	>2.5	>2.5	>2.5	0.625

MIC values for extracts with no antibacterial activity (Inactive; MIC > 10,000 µg/mL), low activity (**2000–5000 µg/mL; shown in bold**), moderate activity (1000–2000 µg/mL; shown in blue), noteworthy activity (400–1000 µg/mL; shown in red), or good activity (100–400 µg/mL; shown in green). Reference antibiotics: PEN G = penicillin G, ERY = erythromycin, TET = tetracycline, CHL = chloramphenicol, CIP = ciprofloxacin, POLB = polymyxin B, OXA = oxacillin, AMX = amoxycillin, GEN = gentamicin, VAN = vancomycin. No antibiotic activity (MIC > 2.5 µg/mL).

**Table 2 antibiotics-13-00654-t002:** ∑FIC values calculated for combinations of *P. niruri* extracts and conventional antibiotics.

Bacteria	Extracts	PEN G	ERY	TET	CHL	CIP	POLB	OXA	AMX	GEN	VAN
** *B. cereus* **	PN-Aq	1.00	2.25	2.25	3.00	1.13	-	0.62	1.00	9.81	2.00
	PN-Me	0.63	3.00	3.00	4.51	1.50	-	2.13	0.63	2.00	2.50
** *S. aureus* **	PN-Aq	0.73	1.48	1.24	-	2.41	-	2.50	0.97	-	1.50
	PN-Me	0.75	0.77	1.25	-	2.44	-	1.50	0.51	-	1.50
**MRSA**	PN-Aq	-	-	1.06	-	2.00	-	-	-	-	1.50
	PN-Me	-	-	1.06	-	1.00	-	-	-	-	0.75
**ESBL *E. coli***	PN-Aq	-	-	-	-	-	-	-	-	-	-
	PN-Me	-	-	-	-	-	63.00	-	-	-	-
** *K. pneumoniae* **	PN-Aq	-	-	1.50	1.50	2.25	63.50	-	-	>4	-
	PN-Me	-	-	1.00	1.25	2.50	32.25	-	-	2.00	-
** *S. sonnei* **	PN-Aq	-	-	1.12	1.00	2.06	63.00	-	-	1.00	-
	PN-Me	-	-	4.53	>4	2.06	>4	-	-	1.00	-
** *S. flexneri* **	PN-Aq	-	-	0.62	0.75	2.06	15.62	-	-	1.25	-
	PN-Me	-	-	0.63	0.75	2.06	15.63	-	-	1.25	-

∑FIC values of *P. niruri* aqueous (PN-Aq) and methanol (PN-Me) extracts in combination with reference antibiotics against antibiotic-sensitive and resistant strains of *B. cereus*, *S. aureus*, MRSA, ESBL *E. coli*, *K. pneumoniae*, *S. sonnei*, and *S. flexneri*. Additive interaction: >0.5≤1.00 (shown in blue); Indifferent interaction: >1.01–≤4.00; Antagonistic interaction: >4.0 (shown in red). - indicates the extract or the antibiotic was inactive against the bacteria being tested.

**Table 3 antibiotics-13-00654-t003:** LC-MS putative identification and % relative abundance of phytochemicals identified in the aqueous (Aq), methanolic (MeOH), and ethyl acetate (EtOAc) plant extracts using negative ionisation mode.

Retention Time [min]	Molecular Mass	Empirical Formula	Putative Compounds	Relative Abundance (% of Total Area)		
				Aq	MeOH	EtOAc
**Isomers**						
1.416	192.02659	C_6_ H_8_ O_7_	Isocitric acid	1.56%	1.07%	-
1.482	116.01078	C_4_ H_4_ O_4_	Maleic acid	0.32%	0.43%	-
1.669	116.01081	C_4_ H_4_ O_4_	Fumaric acid	-	0.10%	-
1.727	131.09453	C_6_ H_13_ N O_2_	Isoleucine	0.14%	0.27%	-
**Organic Compounds**						
1.794	126.03153	C_6_ H_6_ O_3_	Pyrogallol	5.39%	0.92%	-
2.06	166.0629	C_9_ H_10_ O_3_	Apocynin	-	0.04%	-
2.126	170.02141	C_7_ H_6_ O_5_	Gallic acid	11.29%	0.18%	-
2.552	110.03666	C_6_ H_6_ O_2_	Catechol	-	0.05%	-
2.688	154.02651	C_7_ H_6_ O_4_	Protocatechuic acid	1.00%	-	-
2.803	164.0473	C_9_ H_8_ O_3_	4-Coumaric acid	0.30%	1.41%	-
9.019	290.07884	C_15_ H_14_ O_6_	Epicatechin	0.35%	0.59%	-
9.545	288.06304	C_15_ H_12_ O_6_	(-)-Fustin	-	0.13%	-
9.882	302.00597	C_14_ H_6_ O_8_	Ellagic acid	-	0.05%	-
10.575	164.04732	C_9_ H_8_ O_3_	2-Hydroxycinnamic acid	0.59%	1.61%	-
10.639	448.1004	C_21_ H_20_ O_11_	Orientin	-	0.12%	-
10.645	564.14774	C_26_ H_28_ O_14_	Corymboside	-	0.06%	-
10.726	302.04228	C_15_ H_10_ O_7_	2-(2,6-dihydroxyphenyl)-3,5,7-trihydroxy-4H-chromen-4-one	-	0.38%	-
10.726	610.15282	C_27_ H_30_ O_16_	Quercetin 3-O-rhamnoside-7-O-glucoside	-	0.17%	-
10.968	596.13747	C_26_ H_28_ O_16_	2-(3,4-Dihydroxyphenyl)-5,7-dihydroxy-4-oxo-4H-chromen-3-yl 6-O-β-D-xylopyranosyl-β-D-glucopyranoside	0.04%	-	-
10.972	464.09493	C_21_ H_20_ O_12_	Quercetin-3β-D-glucoside	-	0.08%	-
11.134	464.09502	C_21_ H_20_ O_12_	Hyperoside	-	1.69%	0.70%
11.134	448.10015	C_21_ H_20_ O_11_	2-(3,4-dihydroxyphenyl)-5,7-dihydroxy-3-{[(2S,3R,4R,5R,6S)-3,4,5-trihydroxy-6-methyloxan-2-yl]oxy}-4H-chromen-4-one	-	0.23%	-
11.217	432.1052	C_21_ H_20_ O_10_	Vitexin	0.07%	0.32%	-
11.231	478.0744	C_21_ H_18_ O_13_	Miquelianin	1.46%	5.62%	-
11.299	464.09519	C_21_ H_20_ O_12_	Myricitrin	-	0.69%	0.31%
11.367	286.04738	C_15_ H_10_ O_6_	Fisetin	-	0.20%	-
11.5	610.15333	C_27_ H_30_ O_16_	Rutin	-	12.00%	5.85%
11.588	448.10018	C_21_ H_20_ O_11_	Trifolin	-	0.73%	0.98%
11.766	286.04747	C_15_ H_10_ O_6_	Kaempferol	0.11%	2.43%	2.00%
11.783	448.10041	C_21_ H_20_ O_11_	Astragalin	-	0.09%	-
12.198	152.12004	C_10_ H_16_ O	Citral	-	0.04%	-
12.491	302.04225	C_15_ H_10_ O_7_	2-(2,4-dihydroxyphenyl)-3,5,7-trihydroxy-4H-chromen-4-one	-	-	0.24%
12.579	244.13104	C_12_ H_20_ O_5_	3,8,9-trihydroxy-10-propyl-3,4,5,8,9,10-hexahydro-2H-oxecin-2-one (herbarumin II)	0.35%	0.18%	0.65%
13.219	302.04249	C_15_ H_10_ O_7_	Quercetin	0.42%	7.68%	2.37%
13.222	264.13583	C_15_ H_20_ O_4_	Ambrosic acid	-	0.05%	0.44%
14.194	270.05268	C_15_ H_10_ O_5_	Apigenin	-	0.05%	-
17.007	252.17238	C_15_ H_24_ O_3_	Ageratriol	0.26%	-	0.37%
17.685	268.07334	C_16_ H_12_ O_4_	7-hydroxy-3-(4-methoxyphenyl)-4H-chromen-4-one	-	-	0.58%

## Data Availability

Data are either presented within the manuscript or are available from the corresponding author upon reasonable request.

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
