# Peer review of "Phyllanthus niruri Linn.: Antibacterial Activity, Phytochemistry, and Enhanced Antibiotic Combinatorial Strategies"

_antibiotics, 2024, doi:10.3390/antibiotics13070654_

Round 1

Reviewer 1 Report

Comments and Suggestions for Authors

This article can be publish with minor revision

please see attach file

Author Response

Reviewer 1

Reviewer 1 indicated their requested revisions within the manuscript. We have made all corrections that the reviewer highlighted in the revised manuscript, with the exception of the following changes:

  • Line 20: The reviewer has requested Artemia nauplii to be completely italicised. We have not altered this as nauplii is not part of the species name, and therefore should not be italicised. It is a development stage (larvae).
  • Discussion section, comment on line 350. We agree with the reviewer that to highlight any of the identified compounds that have known antibacterial activity is important, and we have already done this. We refer the reviewer to the paragraph following the one they highlighted. We have discussed the activity previously reported for the compounds identified in our study. Therefore, we believe that we have already done as the reviewer has requested. 

In addition, deleting the section of text in the introduction highlighted by the reviewer has resulted in several citations being deleted. These have now been removed from the reference list and the affected intext citations have now been renumbered.

Reviewer 2 Report

Comments and Suggestions for Authors

The manuscript entitled Phyllanthus niruri Linn.: Antibacterial Activity, Phytochemistry and Enhanced Antibiotic Combinatorial Strategies is an interesting work that demonstrates the potential drug from the selected plant with beneficial properties. However, the quality of the manuscript may be strengthened further by addressing the following queries:

1.       In the abstract section, first time used abbreviation or genus names may be expanded for the broader readers, particularly the terms such as FIC, LC-MS, MIC, MRSA, S. aureusK.  pneumoniae, B. cereus, S. flexneri, S. sonnei, P. niruri.

2.       Additionally, the similar expansion of the first time abbreviation used in the manuscript may be done for the better presentation.

3.       Lines 106-116, kindly summarized the patterns of using the similar plant study rather than the discussion of the data values. Similar data values may be discussed during the results and discussion section.

4.       Figure 1 & 2 has six bar charts. Authors should provide the particular name in place of term sample on the x-axis for better understanding of the provided data trends. Similarly, blank samples may be presented in the beginning for the better demarcation of the present study.

5.       Line 224, please mention the name of databases employed for searching the queries/drug in the present study.

6.       In the discussion section, lines 259-262, repetition of beta-lactams derived antibiotics names may be omitted.

7.       Figure 3 scientific names in the caption may be italicized.

8.       In the reference section, scientific names used in the paper title of all the references may be italicized. 

Author Response

Reviewer 2

The manuscript entitled Phyllanthus niruri Linn.: Antibacterial Activity, Phytochemistry and Enhanced Antibiotic Combinatorial Strategies is an interesting work that demonstrates the potential drug from the selected plant with beneficial properties. However, the quality of the manuscript may be strengthened further by addressing the following queries:

  1. In the abstract section, first time used abbreviation or genus names may be expanded for the broader readers, particularly the terms such as FIC, LC-MS, MIC, MRSA,  aureusK.  pneumoniae, B. cereus, S. flexneri, S. sonnei, P. niruri.

All abbreviations have now been explained in the abstract.

  1. Additionally, the similar expansion of the first time abbreviation used in the manuscript may be done for the better presentation.

All abbreviations have now been explained at first mention within the manuscript.

  1. Lines 106-116, kindly summarized the patterns of using the similar plant study rather than the discussion of the data values. Similar data values may be discussed during the results and discussion section.

We agree with the reviewer regarding the value of comparing the trends noted in previous studies. We have also highlighted trends where relevant in the results section (particularly for the solid phase assays, combinational studies, phytochemical identifications and the toxicity studies). However, we acknowledge that we have also included the actual data values when discussing some results, particularly the MIC values. However, we believe that this is very important for these results as the MIC values highlight the potency of the extracts and allow for direct comparisons, both within this study and between studies). Therefore, we believe that it is relevant and important to include these values. Similarly, we have included specific MIC values in the discussion section, which has allowed us to compare the results of our study to other studies. We therefore believe that it is best to not remove these MIC values from the results or discussion sections.

  1. Figure 1 & 2 has six bar charts. Authors should provide the particular name in place of term sample on the x-axis for better understanding of the provided data trends. Similarly, blank samples may be presented in the beginning for the better demarcation of the present study.

We understand the reviewer’s comment and had initially planned to provide a more complete listing for each of the items on the X axes in these figures. However, we found that this made the figures messy and difficult to understand. Therefore, we opted instead for the use of abbreviations, which are then explained below the figures in the legend. We believe that the data is better presented that way and therefore prefer to leave the figures in their currently format.

Additionally, we have two blank/negative controls in in each of these figures. The 1% DMSO serves as a blank for the extracts (which are also in 1% DMSO). Therefore, we placed this before the extract data. The bar marked “blank” is deionised water and serves as the blank for the antibiotic controls, which are also prepared in deionised water. Therefore, we have positioned this blank before the antibiotic controls that they relate to. We believe that this is the best way to display this data.

  1. Line 224, please mention the name of databases employed for searching the queries/drug in the present study.

We agree with the reviewer that naming the databases used is important. However, we do not believe that these should be included where the reviewer has indicated (in the results section). Instead, we believe that the correct place for this methodology detail is in the Materials and Methods section. We have already included details of the databases used in section 4.8. We refer the reviewer to lines 541-548 (and shown below). We therefore believe that this point has already been addressed.

“…databases such as Mz cloud, ChemSpider, Predicted Compositions, and MassList were used to obtain a partial or complete match of the compounds that were found. The Compound Discoverer's Mz cloud and ChemSpider database must pass communication tests to verify online access, accurately identify eluted chemicals, and prevent workflow mistakes. Using the Compound Discoverer software, Excel files of the compounds that were identified from each extract were created. These files were then compared with one another to create a list of potential putative compounds.”

  1. In the discussion section, lines 259-262, repetition of beta-lactams derived antibiotics names may be omitted.

We agree with the reviewer’s comment and have now removed the second listing of these antibiotics on line 262.

  1. Figure 3 scientific names in the caption may be italicized.

We thank the reviewer for pointing this out. P. nururi has now been italicised the revised manuscript.

  1. In the reference section, scientific names used in the paper title of all the references may be italicized. 

We have now italicised scientific names in the reference list where needed.

Reviewer 3 Report

Comments and Suggestions for Authors

The current manuscript can be accepted for publication on condition that the authors respond to the following comments and inquiries. Upon receiving the authors’ response, the manuscript can be accepted for publication.

1.     The novelty of this work is not clear. Please justify the novelty by comparing the following papers

Ibrahim, D.; Hong, L. S.; Kuppan, N. Antimicrobial activity of crude methanolic extract from Phyllanthus niruri. Natural product communications 2013, 8 (4), 1934578X1300800422.

Amin, Z. A.; Abdulla, M. A.; Ali, H. M.; Alshawsh, M. A.; Qadir, S. W. Assessment of in vitro antioxidant, antibacterial and immune activation potentials of aqueous and ethanol extracts of Phyllanthus niruri. J. Sci. Food Agric. 2012, 92 (9), 1874-1877

2.       The authors should provide the LCMS chromatogram of Aqueous and methanol extracts of P. niruri

3.        In Table 2, please provide an additional column for the detected molecular ion (+ve or -ve ), calculated molecular ion, fragment ions, and error in ppm.

4.       It has been demonstrated that the ZOI for E. coli and S. aureus can be more clearly distinguished by method derived by The Jouranal of Antibiotics volume 68pages657–659 (2015). Have you considered a replicate of screening with this method, and also provide the images of plates?

5.       Five flavonoids from P. niruri are PAINS (epicatechin, quercetin, rutin, kaempferol and apigenin). Do you believe observed activity belongs to PAINS?

6.       Why were not all ESKAPE pathogens considered in your assays?

Author Response

Reviewer 3

The current manuscript can be accepted for publication on condition that the authors respond to the following comments and inquiries. Upon receiving the authors’ response, the manuscript can be accepted for publication.

  1. The novelty of this work is not clear. Please justify the novelty by comparing the following papers

Ibrahim, D.; Hong, L. S.; Kuppan, N. Antimicrobial activity of crude methanolic extract from Phyllanthus niruri. Natural product communications 2013, 8 (4), 1934578X1300800422.

Amin, Z. A.; Abdulla, M. A.; Ali, H. M.; Alshawsh, M. A.; Qadir, S. W. Assessment of in vitro antioxidant, antibacterial and immune activation potentials of aqueous and ethanol extracts of Phyllanthus niruri. J. Sci. Food Agric. 2012, 92 (9), 1874-1877

We disagree with the reviewer in the first point in this comment. We believe that the novelty of the study has been clearly stated. We have stated that whilst there are earlier studies examining antibacterial activity of P. niruri extracts the majority of those did not quantify the activity through MIC calculation. This makes comparisons between studies impossible and does not differentiate extracts that are promising and should be further evaluated from those with low activity. Other studies, including the first of the studies listed by the reviewer, report very high MIC values. Indeed, that study reported MIC values between 12.5 and 50 mg/mL for the extracts against the bacteria tested. By most definitions, these very high MICs would actually indicate LACK of activity, or at best, very low activity. Thus, that findings of that study are dubious and required confirmation/repeating. Therefore, there was a need to more accurately test and quantify the potency of P. niruri extracts. Furthermore, as stated in the final paragraph of the introduction, the earlier studies have generally tested against antibiotic-sensitive bacterial strains, whereas our study also screens against resistant strains. Furthermore, our study identifies noteworthy compounds in the bioactive extracts. Therefore, we believe that we have clearly highlighted several differences between our study and the previous studies.

Regarding the comment that we should compare to the studies that the reviewer has highlighted, we refer the reviewer to lines 99-109 where we have not only discussed the Amin et al (2012) study that the reviewer has highlighted, but we have in fact discussed in some detail. We had not included the Ibrahim et al (2013) study because (as discussed in the previous paragraph), we have concerns with the validity of that study. In particular, the very high MIC values reported in that study as good activity highlight interpretation issues. However, to address the reviewers comment, we have now included the following section to the end of the first paragraph on page 3:

Ibrahim and colleagues [12] prepared methanolic extracts of P. niruri, however the process of extraction yielded an oily paste which may have contained traces of methanol solvent. Furthermore, activity on agar was only possible at very high concentrations (100 mg/mL) and high MIC values were determined using microdilution broth assays for B. cereus, B. subtilis and S. aureus (3.13 – 6.25) as well as E. coli, P. aeruginosa and P. rettgeri (25 mg/mL).”

  1. The authors should provide the LCMS chromatogram of Aqueous and methanol extracts of P. niruri

The chromatograms have now been provided as supplementary data, and the text on line 223/224 has been revised to indicate this.

  1. In Table 2, please provide an additional column for the detected molecular ion (+ve or -ve ), calculated molecular ion, fragment ions, and error in ppm.

We thank the reviewer for pointing this out. All of our LC-MS studies used -ve ionisation and we agree that this should have been mentioned in the methods section. To address the reviewers this has now been explained in the title to Table 3, as well as in Supplementary Figure 1. Additionally, Line 537 in the methods section now states that negative ionisation mode was used.

  1. It has been demonstrated that the ZOI for E. coliand S. aureus can be more clearly distinguished by method derived by The Jouranal of Antibiotics volume 68, pages657–659 (2015). Have you considered a replicate of screening with this method, and also provide the images of plates?

We agree that the method that they have highlighted is useful to enhance visibility of ZOIs in some cases. In particular, Figure 2 of the Bhattacharjee 2015 study demonstrates that the staining method is particularly useful for relatively short growth periods. Indeed, the ZOIs are substantially clearer at 4- and 5-hour incubation periods for stained compared to unstained plates. However, it is evident from Figure 3 of that study that the zones are clearly visible both with and without staining after a 24-hour incubation period. Our study also used 24-hour incubation periods and the ZOIs were clear and easy to visualise without staining. Therefore, additional handling using this staining method was not necessary in our study and was not included.

Additionally, we have not provided images of the plates within the manuscript as we believe that is would unnecessarily lengthen the manuscript for no improvement in the manuscript. Indeed, as we tested 3 extracts (as well as 14 controls and conventional antibiotics), all in triplicate against 6 bacteria, this would require the inclusion of >40 photos of agar plates. We believe that it is better to summarise data in the manuscript (as in Figures 1 and 2), rather than present all raw data. However, we are willing to provide any data (including these) to readers upon reasonable request. Additionally, a representative agar plate showing clearly defined ZOIs is already provided in the graphical abstract.

  1. Five flavonoids from P. niruriare PAINS (epicatechin, quercetin, rutin, kaempferol and apigenin). Do you believe observed activity belongs to PAINS?

As highlighted by the reviewer, we believe that those flavonoids may contribute to the activity reported, but may not be solely responsible for that activity. We have already discussed previously studies reporting activity for flavonoids in lines 376-385.

“Flavonoids and their derivatives are widely recognised for their antimicrobial activity. For example, noteworthy antibacterial activity has been reported for quercetin against five clinical strains of MRSA, with MIC values ranging from 16-64 µg/mL [34]. Additionally, dihydromyricetin and myricetin demonstrated strong anti-MRSA activity in the same study (MIC = 10 µg/mL and 25 µg/mL, respectively). Another study reported that flavonoids exert anti-MRSA activity through the modulation of the PBP2a proteins [35]. Buchmann et al. (2022) reported synergistic combinations of myricetin with tetracycline and ciprofloxacin against ESKAPE category K. pneumoniae strains [36]. That study demonstrated that the MIC value of tetracycline against K. pneumoniae (PBIO1455) was decreased by myricetin from 11 μg/mL to 3.3 μg/mL.”

However, the contribution of these compounds to the activities reported in our study need to be confirmed, and this will be a focus of future studies in our group. Additionally, future studies are required to determine whether these compounds can also potentiate the effects of conventional antibiotics, thereby repurposing them (even against bacteria otherwise resistant to their effects). This will also be the focus of future studies. To address the reviewer’s comment, the conclusions section has now been revised to address these points:

The urgent need for new antibacterial medicines has sparked interest in natural products as possible sources. Our findings indicate that active extracts of Phyllanthus niruri L. can inhibit the growth of resistant bacteria just as efficiently as susceptible strains, indicating distinct antibacterial mechanisms for the extracts. Furthermore, the methanolic and aqueous extracts potentiate the activity of several conventional antibiotics, particularly, β-lactam antibiotics. This may allow these antibiotics to be reactivated, even in bacterial strains otherwise resistant to their effects. The potentiation mechanism is yet to be determined, but it is possible that extract components may inactivate bacterial extended spectrum β-lactamase enzymes, thereby increasing the intracellular antibiotic concentration. Some of the phytochemicals identified are likely to contribute to these activities, implying that they could be useful as novel antibacterial agents, although this needs to be verified. Future research should focus on isolating these molecules and investigating their potential to enhance the efficacy of current antibiotics. The use of additional methods such as GC-MS are needed to identify lower polarity and volatile molecules, whilst NMR is required to definitively structurally identify the extract components.”

  1. Why were not all ESKAPE pathogens considered in your assays?

The focus of this study was to examine antibacterial activity, with a focus on antibiotic resistant bacteria. For this reason, we chose to examine resistant and sensitive pairs of E. coli, S. aureus and K. pneumoniae. For each of those bacteria, we had access to the both members of the pairs (resistant and sensitive). Thus, whilst this project screened against 3 of the ESKAPE bacteria, we didn’t screen against the others due to lack of access. To address our inability to screen against the other ESKAPE bacteria, the follow text has been added to lines 396-398:

“Further studies are required to screen the extracts against the other ESKAPE bacteria to more fully evaluate their potential against these pathogens.”

Reviewer 4 Report

Comments and Suggestions for Authors

The manuscript by Tiwana et al. describes the study on the antibacterial activities of three Phyllanthus niruri extracts (methanol, water, and ethyl acetate). Extracts were characterized by LC-MS. Combination assays were also performed for aqueous and methanol extracts, but no synergistic interactions were observed. Additionally, extracts showed low toxicity according to the Artemia toxicity assay.

1) The major issue of the paper is the chemical characterization of the extracts (Section 2.3 and Table 3). Only a small fraction of the extracts was identified—the total percentage of the identified constituents was under 24% for the H2O extract, and even less for the EtOAc extract (15%). Most of the identified compounds have abundances < 1%, and I assume that the major constituents were left unidentified. I strongly recommend that the authors perform additional characterization and attempt to identify the major constituents (for example, through preparative chromatography and NMR). Additionally, GC-MS analysis could provide a lot of data on the volatile components from the EtOAc extract.

2) There is no reason to include the 'Molecular Mass' and 'Empirical Formula' columns in Table 3, as this data can be easily looked up.

3) The authors should provide the chromatograms of the extracts as part of the Supporting Information file.

4) If possible, the authors should quantify the constituents with the highest % content from Table 3 (pyrogallol, gallic acid, rutin, quercetin).

5) The authors reported that the concentration of the stock solutions in the Artemia assay was 2 mg/mL (line 506). Did the EtOAc extract fully dissolve in artificial seawater? This concentration is quite high, and I assume that the EtOAc extract contained a lot of non-polar compounds.

6) There is already a lot of data on the antibacterial activities of P. niruri leaf extracts in the literature. The authors should clearly explain the novelty of their work.

Author Response

Reviewer 4

  • The major issue of the paper is the chemical characterization of the extracts (Section 2.3 and Table 3). Only a small fraction of the extracts was identified—the total percentage of the identified constituents was under 24% for the H2O extract, and even less for the EtOAc extract (15%). Most of the identified compounds have abundances < 1%, and I assume that the major constituents were left unidentified. I strongly recommend that the authors perform additional characterization and attempt to identify the major constituents (for example, through preparative chromatography and NMR). Additionally, GC-MS analysis could provide a lot of data on the volatile components from the EtOAc extract.

The reviewer is correct – the compounds listed in Table 3 are only a portion of the compounds in the extracts (<25% as the reviewer has stated. We had used metabolomic profiling to identify as many compounds as possible, although we chose not to include all in our manuscript as we believed that the large number of compounds identified (which resulted in a long table) unnecessarily lengthens the manuscript. Instead, we narrowed the focus to only the compounds (particularly flavonoids and tannins) that we believed are more likely to contribute to the activities reported in our study for the sake of brevity. These classes of compounds were selected because of the wealth of information of flavonoids and tannins with antibacterial activity. This has been discussed in the discussion section.

However, we understand the reviewer’s point and we agree that the complete list of identified compounds may be useful, although we still believe that this would lengthen the manuscript unnecessarily. Therefore, to address the reviewer’s comment, we have included this information as Supplementary Table 1.

  • There is no reason to include the 'Molecular Mass' and 'Empirical Formula' columns in Table 3, as this data can be easily looked up.

Whilst we agree with the reviewer that it is not necessary to provide this information and that the readers can look this up, these columns do not add to the length of the manuscript. They make it easier for readers and therefore, whilst including this information may not be absolutely necessary, we believe that it does increase the quality of the manuscript and we have left the column in the table.

  • The authors should provide the chromatograms of the extracts as part of the Supporting Information file.

The chromatograms have now been provided as supplementary data, and the text on line 223/224 has been revised to indicate this.

  • If possible, the authors should quantify the constituents with the highest % content from Table 3 (pyrogallol, gallic acid, rutin, quercetin).

We had decided to present our data as a fingerprint analysis, focussing on the flavonoid and tannin components due to the antibacterial properties of these classes of compounds. The compounds presented in Table 3 represent only compounds from those phytochemical classes, and this has been stated in the text. However, we understand the reviewer’s comment and have also included the list of other compounds as Supplementary Table 1. The relative % abundances of all identified compounds have been included.

  • The authors reported that the concentration of the stock solutions in the Artemia assay was 2 mg/mL (line 506). Did the EtOAc extract fully dissolve in artificial seawater? This concentration is quite high, and I assume that the EtOAc extract contained a lot of non-polar compounds.

The reviewer has misunderstood the method. The test solutions were prepared at 2mg/mL concentrations, which were then added to equal volumes of artificial seawater in the assay (lines 511-513):

“Briefly, 400 µL of plant extracts (2mg/mL diluted in artificial seawater) and 400 µL of artificial seawater containing newly hatched (within 1 day) A. franciscana nauplii were dispensed into individual wells of a 48 well plate.”

 Thus, the extracts were actually tested at 1 mg/mL concentrations in the assay, rather than 2 mg/mL. Notably, ethyl acetate extracted only low masses of material (compared to the aqueous and methanolic extracts), which is one of the reasons why we selected the low concentration of 1 mg/mL as the highest concentration to test in this assay (also, LC50 values >1mg/mL are defined as nontoxic in this assay). All of the material in the extracts dissolved in this system. This was achieved by initially using a small volume of 100% DMSO to aid in dissolving the low polarity compounds. The volume was then increased to the final volume, resulting in 1% DMSO in the final solution. The relevant method is provided already in section 4.2 and is shown below:

“All dried extracts were weighed to determine the final yields and then resuspended in 10 mL 1% dimethyl sulfoxide (DMSO; Merck, Australia) and sonicated three times using 20 seconds pulses of a probe sonicator set at 1kHz, with 30 seconds rest between pulses.”

The extracted material remained in solution as the extract volume was increased to the final concentration, and as the test solution was added to the artificial seawater.

  • There is already a lot of data on the antibacterial activities of  nirurileaf extracts in the literature. The authors should clearly explain the novelty of their work.

We agree with the reviewer that there is some literature available on the antibacterial activities of P. niruri leaf extracts in the literature, although we disagree that there is a lot of data, and much of the data published is either inadequate or dubious. We have already discussed this within the manuscript. Additionally, another reviewer also raised the same point and recommended that we reference the publications below and discuss them. As the response in the same, we have copied our response to the other reviewer below:  

Ibrahim, D.; Hong, L. S.; Kuppan, N. Antimicrobial activity of crude methanolic extract from Phyllanthus niruri. Natural product communications 2013, 8 (4), 1934578X1300800422.

Amin, Z. A.; Abdulla, M. A.; Ali, H. M.; Alshawsh, M. A.; Qadir, S. W. Assessment of in vitro antioxidant, antibacterial and immune activation potentials of aqueous and ethanol extracts of Phyllanthus niruri. J. Sci. Food Agric. 2012, 92 (9), 1874-1877

We disagree with the reviewer on this point. We believe that the novelty of the study has been clearly stated. We have stated that whilst there are earlier studies examining antibacterial activity of P. niruri extracts the majority of those did not quantify the activity through MIC calculation. This makes comparisons between studies impossible and does not differentiate extracts that are promising and should be further evaluated from those with low activity. Other studies, including the first of the studies listed by the reviewer, report very high MIC values. Indeed, that study reported MIC values between 12.5 and 50 mg/mL for the extracts against the bacteria tested. By most definitions, these very high MICs would actually indicate LACK of activity, or at best, very low activity. Thus, that findings of that study are dubious and required confirmation/repeating. Therefore, there was a need to more accurately test and quantify the potency of P. niruri extracts. Furthermore, as stated in the final paragraph of the introduction, the earlier studies have generally tested against antibiotic-sensitive bacterial strains, whereas our study also screens against resistant strains. Furthermore, our study identifies noteworthy compounds in the bioactive extracts. Therefore, we believe that we have clearly highlighted several differences between our study and the previous studies.

Regarding the comment that we should compare to the studies that the reviewer has highlighted, we refer the reviewer to lines 99-109 where we have not only discussed the Amin et al (2012) study that the reviewer has highlighted, but we have in fact discussed in some detail. We had not included the Ibrahim et al (2013) study because (as discussed in the previous paragraph), we have concerns with the validity of that study. In particular, the very high MIC values reported in that study as good activity highlight interpretation issues. However, to address the reviewer’s comment, we have now included the following section to the end of the first paragraph on page 3:

Ibrahim and colleagues [12] prepared methanolic extracts of P. niruri, however the process of extraction yielded an oily paste which may have contained traces of methanol solvent. Furthermore, activity on agar was only possible at very high concentrations (100 mg/mL) and high MIC values were determined using microdilution broth assays for B. cereus, B. subtilis and S. aureus (3.13 – 6.25 mg/mL) as well as E. coli, P. aeruginosa and P. rettgeri (25 mg/mL).”

Reviewer 5 Report

Comments and Suggestions for Authors

In this manuscript, Gagan Tiwana et al. studied the antibacterial properties of extracts from Phyllanthus niruri Linn. against various pathogens. They found significant antibacterial activity against S. aureus, MRSA, K. pneumoniae, and B. cereus using microdilution assays. LC-MS analysis identified flavonoids and tannins in the extracts. The study suggests these extracts could be valuable for antibiotic development, with further research needed on their mechanisms and phytochemistry.

Overall, the manuscript is well-written. The introduction sufficiently provides background information to elucidate the significance of this study. The study design adheres to a traditional approach in natural product discovery from plant extracts. The results are particularly promising, indicating significant antibacterial activity against several strains. The presence of multiple bioactive components within the extracts stimulates interest in further research aimed at antibiotic discovery.

I recommend accepting in the present form of this manuscript.

Author Response

Reviewer 5

In this manuscript, Gagan Tiwana et al. studied the antibacterial properties of extracts from Phyllanthus niruri Linn. against various pathogens. They found significant antibacterial activity against S. aureus, MRSA, K. pneumoniae, and B. cereus using microdilution assays. LC-MS analysis identified flavonoids and tannins in the extracts. The study suggests these extracts could be valuable for antibiotic development, with further research needed on their mechanisms and phytochemistry.

 Overall, the manuscript is well-written. The introduction sufficiently provides background information to elucidate the significance of this study. The study design adheres to a traditional approach in natural product discovery from plant extracts. The results are particularly promising, indicating significant antibacterial activity against several strains. The presence of multiple bioactive components within the extracts stimulates interest in further research aimed at antibiotic discovery.

I recommend accepting in the present form of this manuscript.

We thank the reviewer for the positive feedback. No changes have been incorporated in response to this review.

Round 2

Reviewer 3 Report

Comments and Suggestions for Authors

The current manuscript can be accepted for publication on condition that the authors respond to the following comments and inquiries. Upon receiving the authors’ response, the manuscript can be accepted for publication.

Figure 3 does indicate that ZOIs may be clearly visible with and without staining, but the work from Bhattacharjee clearly shows that agar volume in solid-based antimicrobial susceptibility assays can significantly affect the clarity of the ZOIs. Flanagan and Steck (Indian Journal of Microbiology. 2017 Dec; 57(4): 503-506.) also reported that agar depth and uniformity affect the ZOI in antimicrobial susceptibility assays. This work highlights the role of diffusion rate the efficacy of tested antimicrobials: “…the higher the agar diffusion rate for an agent, the more susceptible is the resulting MIC and zone of inhibition value to agar depth.” Not only could this be the cause of the discrepancies between your agar and broth results. Also, did you verify that all agar volumes were of the same depth and uniformity?

Author Response

Reviewer 3

The current manuscript can be accepted for publication on condition that the authors respond to the following comments and inquiries. Upon receiving the authors’ response, the manuscript can be accepted for publication.

Figure 3 does indicate that ZOIs may be clearly visible with and without staining, but the work from Bhattacharjee clearly shows that agar volume in solid-based antimicrobial susceptibility assays can significantly affect the clarity of the ZOIs. Flanagan and Steck (Indian Journal of Microbiology. 2017 Dec; 57(4): 503-506.) also reported that agar depth and uniformity affect the ZOI in antimicrobial susceptibility assays. This work highlights the role of diffusion rate the efficacy of tested antimicrobials: “…the higher the agar diffusion rate for an agent, the more susceptible is the resulting MIC and zone of inhibition value to agar depth.” Not only could this be the cause of the discrepancies between your agar and broth results. Also, did you verify that all agar volumes were of the same depth and uniformity?

We have now added statements within the Discussion covering these aspects at line 246:

A previous study has provided evidence that agar depth in the petri dishes as well as agar uniformity can affect the size of ZOIs in the agar disc diffusion assays [19]. Although we prepared uniform agar according to manufacturers’ instructions and poured the agar at a consistent depth of 4 mm, we acknowledge that the disc diffusion assays should be repeated at different agar depths in order to verify the ZOIs of all of the extracts.”

As such, the Flanagan and Steck manuscript has been added to the reference list, and the references re-numbered within the text as well as in the reference list.

This is immediately followed by a statement regarding the agar plate staining method mentioned earlier by this reviewer. This text is as follows:

Additionally, while the ZOI boundaries were clearly visible in our study, it should be noted that staining plates with methylene blue or crystal violet [20] may further enhance the visibility of ZOIs should lack of clarity in this regard occur in future studies.”

Accordingly, the Bhattacharjee manuscript has been added to the reference list.

Reviewer 4 Report

Comments and Suggestions for Authors

I would like to thank the authors for addressing the points raised in the review. I have just one additional question: why did the authors include the chromatograms of Terminalia phanerophlebia extracts in the Supplementary Information (Supp. figure S2)? Was there a specific reason for this inclusion?

Author Response

Reviewer 4

I would like to thank the authors for addressing the points raised in the review. I have just one additional question: why did the authors include the chromatograms of Terminalia phanerophlebia extracts in the Supplementary Information (Supp. figure S2)? Was there a specific reason for this inclusion?

This was done for brevity. We do not believe that its inclusion in the manuscript itself adds anything, but we do agree that it is important to make it available for any reader who may wish to see it.